# Novel Modeling Approach to Analyze Threshold Voltage Variability in Short Gate-Length (15–22 nm) Nanowire FETs with Various Channel Diameters

**DOI:** 10.3390/nano12101721

**Published:** 2022-05-18

**Authors:** Seunghwan Lee, Jun-Sik Yoon, Junjong Lee, Jinsu Jeong, Hyeok Yun, Jaewan Lim, Sanguk Lee, Rock-Hyun Baek

**Affiliations:** The Department of Electrical Engineering, Pohang University of Science and Technology (POSTECH), Pohang 37673, Gyeongbuk, Korea; sh5233@postech.ac.kr (S.L.); junsikyoon@postech.ac.kr (J.-S.Y.); lee1539@postech.ac.kr (J.L.); js.jeong@postech.ac.kr (J.J.); myska315@postech.ac.kr (H.Y.); jaewan94@postech.ac.kr (J.L.); sanguk96@postech.ac.kr (S.L.)

**Keywords:** variability modeling, threshold voltage, SNWFET, ultrashort gate-length, Pelgrom’s law, nanowire diameter, metal gate granularity, dopant diffusion

## Abstract

In this study, threshold voltage (*V*_th_) variability was investigated in silicon nanowire field-effect transistors (SNWFETs) with short gate-lengths of 15–22 nm and various channel diameters (*D*_NW_) of 7, 9, and 12 nm. Linear slope and nonzero y-intercept were observed in a Pelgrom plot of the standard deviation of *V*_th_ (σ*V*_th_), which originated from random and process variations. Interestingly, the slope and y-intercept differed for each *D*_NW_, and σ*V*_th_ was the smallest at a median *D*_NW_ of 9 nm. To analyze the observed *D*_NW_ tendency of σ*V*_th_, a novel modeling approach based on the error propagation law was proposed. The contribution of gate-metal work function, channel dopant concentration (*N*_ch_), and *D*_NW_ variations (WFV, ∆*N*_ch_, and ∆*D*_NW_) to σ*V*_th_ were evaluated by directly fitting the developed model to measured σ*V*_th_. As a result, WFV induced by metal gate granularity increased as channel area increases, and the slope of WFV in Pelgrom plot is similar to that of σ*V*_th_. As *D*_NW_ decreased, SNWFETs became robust to ∆*N*_ch_ but vulnerable to ∆*D*_NW_. Consequently, the contribution of ∆*D*_NW_, WFV, and ∆*N*_ch_ is dominant at *D*_NW_ of 7 nm, 9 nm, and 12, respectively. The proposed model enables the quantifying of the contribution of various variation sources of *V*_th_ variation, and it is applicable to all SNWFETs with various *L*_G_ and *D*_NW_.

## 1. Introduction

Gate-all-around (GAA) silicon nanowire field-effect transistors (SNWFETs) are considered as a viable option for future device architecture due to their adequate gate-controllability with GAA structures [1,2,3]. However, ultrascaled SNWFETs suffer from severe threshold voltage (*V*_th_) variation because the device-to-device variation increases with the decrease in the effective channel width (*W*_eff_) and gate-length (*L*_G_) [4,5]. According to previous studies, random variations such as metal gate granularity (MGG), line edge roughness (LER), and random dopant fluctuation (RDF) cause *V*_th_ variation in ultrascaled GAA transistors [6,7,8,9,10]. Additionally, the *V*_th_ variation is also induced by the process variations such as junction gradient and channel thickness variation [11,12,13,14,15,16,17].

Therefore, several simulations and models have been recommended to analyze the contribution of multiple sources to *V*_th_ variation. First, technology computer-aided design (TCAD) simulations are suitable for analyzing the influence of variation sources, but it is difficult to predict the cause of variation inversely from measured *V*_th_ variation [6,7,8,9,10]. Second, simulation program with integrated circuit emphasis (SPICE)-based models can be applied to analyze the variation sources of measured *V*_th_ variation, but it consumes time and makes an error because all devices should be calibrated [15]. Last, models based on the error propagation law have been proposed [16,17]. These modeling approaches enable extraction of the contribution of each variation source to the standard deviation of *V*_th_ (σ*V*_th_) fast and accurately because they directly model σ*V*_th_. However, the error propagation law-based model to analyze the *V*_th_ variability of SNWFET has not been suggested.

Previously, *V*_th_ variability in SNWFETs was investigated considering various *L*_G_ using a SPICE-based model [15]. However, the study did not consider the effect of channel dopant concentration (*N*_ch_) variation and nanowire diameter (*D*_NW_) change. Furthermore, although *D*_NW_ influences *W*_eff_, electrostatics, and quantum effect [18], the *D*_NW_ tendency of *V*_th_ variability in SNWFET with short *L*_G_ has not been thoroughly investigated.

Therefore, in this study, we quantitatively analyzed the sources of *V*_th_ variation in SNWFETs with short *L*_G_ (15–22 nm) and multiple *D*_NW_ (7, 9, 12 nm). A novel modeling approach based on the error propagation law is proposed to estimate the contribution of multiple variation sources to the *V*_th_ variability. The dominant variation source of *V*_th_ variation is analyzed for each *D*_NW_ by using the proposed model. Additionally, the standard deviation of *N*_ch_ (σ*N*_ch_) and *D*_NW_ (σ*D*_NW_) according to *L*_G_ is presented.

## 2. Device Structure and Modeling Methods

### 2.1. Structure and Possible V_th_ Variation Sources of SNWFETs

Figure 1 depicts the schematic and *V*_th_ variation sources of SNWFETs, fabricated using the same process flow reported in [19,20]. The SNWFETs adopted Mid-gap TiN metal gate, gate oxide thickness (*t*_ox_) of 3.4 nm, and (110) channel direction. The gate and nanowire trimming process was used to obtain *L*_G_ varying from 15 to 22 nm and *D*_NW_ of 7, 9, and 12 nm. In this process, *D*_NW_ variation (∆*D*_NW_) was caused by LER occurred at the nanowire (NW) edges and under- or over-etching of the NW [21]. MGG occurred in the TiN metal gate and generated the metal work function variation (WFV) [22]. The transmission electron microscope (TEM) image shows many grain boundaries exist in the TiN metal gate of the SNWFETs [19]. Although the SNWFET is fabricated with an undoped channel, the source/drain (S/D) dopants diffuse into the channel with short *L*_G_ [18,23]. Consequently, *N*_ch_ variation (∆*N*_ch_) was caused by RDF, the S/D dopant implant, annealing, and SiGe strain variation [8,9,10].

Figure 2a depicts the *I*_D_–*V*_G_ characteristics of SNWFETs with *L*_G_ = 15 nm and *D*_NW_ = 7 nm at a drain bias of 0.05 V. About 50 samples were measured per device condition. Here, *V*_th_ was extracted at *I*_D_ = 10^−7^ × *πD*_NW_/*L*_G_ using the constant current method. The fluctuation of extracted *V*_th_ shows the process and random variation affect the physical characteristics of SNWFETs. Figure 2b illustrates a quantile plot of *V*_th_ for each *D*_NW_ in SNWFETs with an *L*_G_ of 15 nm, which shows the distribution of *V*_th_. The distribution of *V*_th_ predominantly follows the theoretical normal distribution for all device conditions, which indicates that sufficient *V*_th_ values were obtained to analyze σ*V*_th_.

Figure 3 is the Pelgrom plot of σ*V*_th_ in SNWFETs for each *D*_NW_, showing the trend of σ*V*_th_ as channel area changes. The slope of the Pelgrom plot, defined as the Pelgrom coefficient (*A*_vt_), represents the effect of random variation [4]. The y-intercept of the Pelgrom plot is also observed, indicating the effect of the process variation and short channel effect [12,24]. Remarkably, the values of *A*_vt_ and y-intercepts differed for each *D*_NW_, and the σ*V*_th_ is smallest in median *D*_NW_ of 9 nm. We anticipated that this result implies a trade-off relationship between the various variation sources. Hence, a novel modeling approach is proposed to analyze the contribution of each variation source to σ*V*_th_.

### 2.2. Proposed σV_th_ Model of SNWFETs

Figure 4 shows the proposed modeling flow to analyze the contribution of WFV, ∆*N*_ch_, and ∆*D*_NW_ to σ*V*_th_. To model σ*V*_th_, we started from a physical model for *V*_th_ of SNWFET, as follows [25,26]:(1)$$V_{\mathrm{th}}=\Phi_{M}-\Phi_{S}-qN_{\mathrm{ch}}\left(\frac{\pi r_{\mathrm{nw}}^{2}}{C_{\mathrm{ox}}}+\frac{r_{\mathrm{nw}}^{2}}{4\varepsilon_{\mathrm{si}}}\right)+\frac{h^{2}}{4\pi m^{*}qr_{\mathrm{nw}}^{2}},$$
where *Φ*_M_ denotes the work function of the TiN gate metal; *Φ*_S_ represents the work function of silicon channel calculated as χ_si_ − *E*_g_/2 + *kT*/*q*∙ln(*N*_ch_/*n*_i_), where χ_si_ is the electron affinity and *E*_g_ is the band gap of silicon; *r*_nw_ indicates the radius of NW; $\varepsilon_{\mathrm{si}}$ and $\varepsilon_{\mathrm{ox}}$ represent the dielectric constant of silicon and oxide, respectively; *h* denotes the Planck constant; and *m** indicates the effective mass of an electron. *C*_ox_ represents the oxide capacitance calculated as 2*πε*_ox_/ln(1 + *t*_ox_/*r*_nw_). The possible *V*_th_ variation sources in Equation (1) are *Φ*_M,_ *N*_ch_, *D*_NW_, and *t*_ox_ variations. Among them, *t*_ox_ is not considered because its variation and effect are very small and negligible [11,12,27]. Although the variation of effective channel length (*L*_eff_) is not considered directly, *N*_ch_ variation partially represents *L*_eff_ variation because S/D dopant diffusion and *L*_G_ variation change *N*_ch_ and *L*_eff_ simultaneously.

Hence, considering three identical variation sources of WFV, ∆*N*_ch_, and ∆*D*_NW_, σ*V*_th_ can be expressed based on the error propagation law as
(2)$$\sigma V_{\mathrm{th}}^{2}=\sigma \Phi_{M}{}^{2}+{\left(\frac{\partial V_{\mathrm{th}}}{\partial N_{\mathrm{ch}}}\cdot \sigma N_{\mathrm{ch}}\right)}^{2}+{\left(\frac{\partial V_{\mathrm{th}}}{\partial D_{\mathrm{NW}}}\cdot \sigma D_{\mathrm{NW}}\right)}^{2}.$$

To analyze σ*V*_th_ using Equation (2), the sensitivity of *V*_th_ against variation sources and their standard deviation should be extracted. First, the standard deviation of metal work function (σ*Φ*_M_) can be estimated by the existing WFV model for SNWFETs, as follows [22]:(3)$$\sigma \Phi_{M}=RGG\times SL=\frac{G_{\mathrm{size}}}{\sqrt{L_{G}\left(D_{\mathrm{NW}}+2t_{\mathrm{ox}}\right)\pi }}\times SL,$$
where *RGG* is the ratio of average grain size to the gate area, *SL* is the sensitivity of σ*V*_th_ against *RGG*, and *G_size_* is the grain size of the metal gate. Here, *G*_size_ can be estimated from a TEM image of the TiN metal gate of SNWFETs. *SL* of SNWFETs can be obtained from previous research based on TCAD simulation [22].

Second, the sensitivity of *V*_th_ against ∆*N*_ch_ and ∆*D*_NW_ can be obtained by calculating the partial differentiation of Equation (1), as follows:(4)$$\frac{\partial V_{\mathrm{th}}}{\partial N_{\mathrm{ch}}}=\frac{kT/q}{N_{\mathrm{ch}}}-qr_{\mathrm{nw}}^{2}\left(\frac{\ln(1+t_{\mathrm{ox}}/r_{\mathrm{nw}})}{2\varepsilon_{\mathrm{ox}}}+\frac{1}{4\varepsilon_{\mathrm{si}}}\right),$$
(5)$$\frac{\partial V_{\mathrm{th}}}{\partial D_{\mathrm{NW}}}=-\frac{qN_{\mathrm{ch}}r_{\mathrm{nw}}}{2}\left(\frac{1}{\varepsilon_{\mathrm{ox}}}\left(2\ln\left(1+\frac{t_{\mathrm{ox}}}{r_{\mathrm{nw}}}\right)-\frac{t_{\mathrm{ox}}}{t_{\mathrm{ox}}+r_{\mathrm{nw}}}\right)+\frac{1}{\varepsilon_{\mathrm{si}}}\right)-\frac{h^{2}}{2\pi m^{*}qr_{\mathrm{nw}}^{3}},$$
where *k* denotes the Boltzmann constant. Here, *N*_ch_ can be extracted where Equation (1) best fits to measured *V*_th_. Finally, σ*N*_ch_ and σ*D*_NW_ are extracted when Equation (2) best fits the square of the measured σ*V*_th_.

The proposed model obtains the *V*_th_ sensitivity against ∆*N*_ch_ and ∆*D*_NW_ through simple calculation and extracts the standard deviation of each variation source by fitting the model to the measured σ*V*_th_. Therefore, the contribution of multiple variation sources to σ*V*_th_ can be directly and quickly modeled and analyzed using the proposed model without any TCAD or SPICE simulation. Furthermore, the proposed *V*_th_ modeling flow is expected to be applied to analyze σ*V*_th_ in most multigate devices with various *L*_G_ and channel thicknesses.

## 3. Results and Discussion

### 3.1. V_th_ Modeling Results of SNWFETs

*N*_ch_ is extracted where Equation (1) fitted *V*_th_ versus *D*_NW_ with high accuracy in SNWFET with *L*_G_ of 15 nm, as shown in Figure 5a. Figure 5b shows *N*_ch_ increases as *L*_G_ decreases because more dopant diffused to the center of the channel from S/D even with the same S/D junction gradient. The sensitivity of *V*_th_ against ∆*N*_ch_ and ∆*D*_NW_ was calculated by substituting *N*_ch_ and other parameters in Equations (4) and (5).

### 3.2. V_th_ Standard Deviation Modeling Results of SNWFETs

#### 3.2.1. Extraction of *G*_size_ and WFV of SNWFETs

*G*_size_ should be determined from the TEM image of the TiN metal gate of the SNWFET to analyze WFV. Figure 6a shows a schematic of the grain boundaries based on TiN metal gate TEM image [19]. *G*_size_ was measured as the average of values obtained by dividing the length of TiN metal in the TEM image (*L*_TEM_) by the number of intersections between grain boundaries and horizontal lines (*N*_int_), as follow [28]:(6)$$G_{\mathrm{size}}=\frac{\sum_{i=1}^{N_{\mathrm{line}}}L_{\mathrm{TEM}}/(N_{\mathrm{int},\;i}+1)}{N_{\mathrm{line}}},$$
where *N*_line_ is the number of horizontal lines. The distance between the lines was set to 5 nm, as shown in Figure 6a. Consequently, *G*_size_ measured using Equation (6) was 11.8 nm in SNWFETs. According to previous research, the value of SL is 105 V/nm in SNWFETs [22]. σ*Φ*_M_ was calculated by putting obtained *G*_size_ and SL into Equation (3), and Figure 6b shows σ*Φ*_M_ as a function of the square root of the channel area. Interestingly, the trend and value of the slope in Figure 6b (*A*_WFV_) are very similar to *A*_vt_ for each *D*_NW_. It means WFV induced by MGG is the dominant random variation component of *V*_th_ variation in the SNWFETs.

#### 3.2.2. The Contribution of Variation Sources to σ*V*_th_ for Each *D*_NW_

Figure 7 shows that Equation (2) accurately fitted the measured σ*V*_th_ with the relative root mean square error of 0.3% where σ*N*_ch_ = 1.11 × 10^18^ cm and σ*D*_NW_ = 0.743 nm. WFV and *D*_NW_ are slightly correlated because *D*_NW_ is included in Equation (3), which can affect the modeling accuracy. However, assuming the occurrence of ∆*D*_NW_ of 0.743 nm, the possible WFV fluctuation is only by 2.4% of total σ*V*_th_ and does not change the *D*_NW_ tendency of σ*V*_th_ induced by each variation source. The *D*_NW_ tendency of σ*V*_th_ can be explained by the different contributions of the three variation sources, which are represented using pink (WFV), red (∆*N*_ch_), and green (∆*D*_NW_) lines in Figure 7. The modeling results are shown considering *L*_G_ = 15 nm; however, the model was also applied to SNWFET with other *L*_G_, and the modeling accuracy and trend of each variation sources are very similar.

Although *A*_MGG_ decreases when *D*_NW_ decreases, the contribution of WFV increases owing to the decrease in the channel area. As *D*_NW_ decreases, SNWFETs become robust to ∆*N*_ch_-induced *V*_th_ variation. This is because the influence of depletion charge and surface potential is reduced proportional to $r_{\mathrm{nw}}^{2}$ because of the improvement in gate-controllability, as shown in Equation (4). Conversely, SNWFETs become vulnerable to ∆*D*_NW_-induced *V*_th_ variation because the sensitivity of *V*_th_ to quantum effect is proportional to 1/$r_{\mathrm{nw}}^{3}$, as indicated in Equation (5). Consequently, the contribution of ∆*D*_NW_, WFV, and ∆*N*_ch_ is dominant when at *D*_NW_ of 7 nm, 9 nm, and 12, respectively.

#### 3.2.3. The Tendency of σ*N*_ch_ and σ*D*_NW_ as *L*_G_ Changes

Figure 8 shows both σ*N*_ch_ and σ*D*_NW_ increases as *L*_G_ decreases. This result means that RDF and LER occur because their influence increases as the device dimension decreases. However, the degree of σ*N*_ch_ and σ*D*_NW_ increase is small, about 5%, as *L*_G_ decreases from 22 to 15 nm. In addition, we already verified WFV by MGG is the dominant random variation component of V_th_ variation in Section 3.2.1. Hence, most ∆*N*_ch_ and ∆*D*_NW_ originated from process variation sources, which causes non-zero y-intercept in Figure 3.

## 4. Conclusions

The contribution of WFV, ∆*N*_ch_, and ∆*D*_NW_ in *V*_th_ variation of SNWFET was quantitatively analyzed for each *D*_NW_ using the novel modeling approach. The sensitivity of WFV against the channel area is similar to that of σ*V*_th_. As *D*_NW_ decreases, SNWFETs became robust to ∆*N*_ch_ but vulnerable to ∆*D*_NW_. The dominant variation sources differed for each *D*_NW_. Hence, the strategy to improve the variability of SNWFETs should be different for each *D*_NW_. Furthermore, with slight modifications, the proposed modeling approach and results are expected to be used in most multigate devices, including FinFET and nanosheet FET.

## Figures and Tables

**Figure 1 nanomaterials-12-01721-f001:**
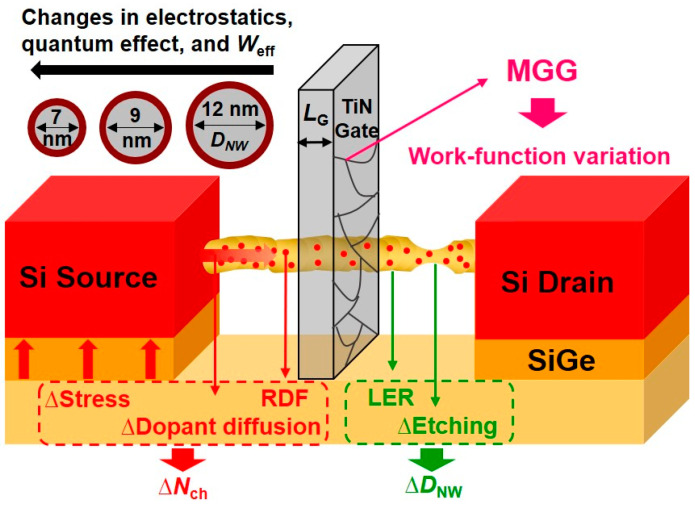
Schematic of the silicon nanowire field-effect transistor (SNWFET) and possible *V*_th_ variation sources.

**Figure 2 nanomaterials-12-01721-f002:**
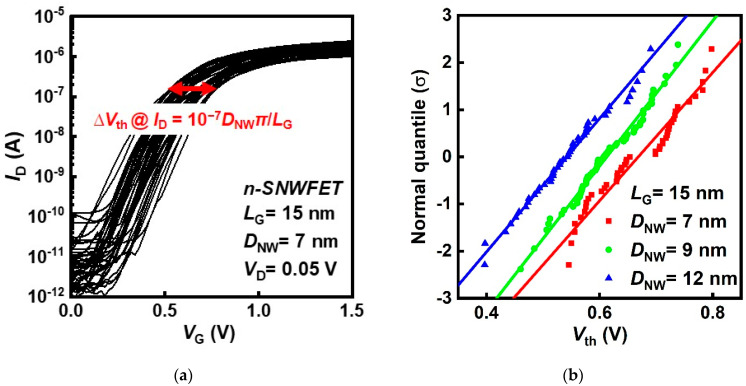
(**a**) *V*_th_ fluctuation in *I*_D_−*V*_G_ of silicon nanowire field-effect transistors (SNWFETs) with *L*_G_ = 15 nm and *D*_NW_ = 7 nm. *V*_th_ is directly extracted using the constant current method at *I*_D_ = 10^−7^∙*πD*_NW_/*L*_G_. (**b**) Quantile plot of *V*_th_ of the SNWFET with *L*_G_ = 15 nm.

**Figure 3 nanomaterials-12-01721-f003:**
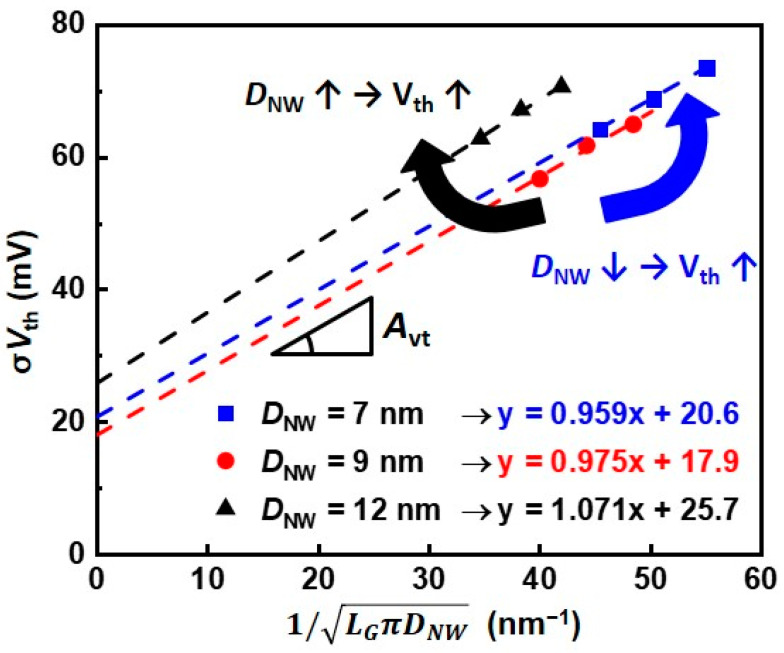
Pelgrom plot for *V*_th_ variation of the silicon nanowire field−effect transistors (SNWFETs).

**Figure 4 nanomaterials-12-01721-f004:**
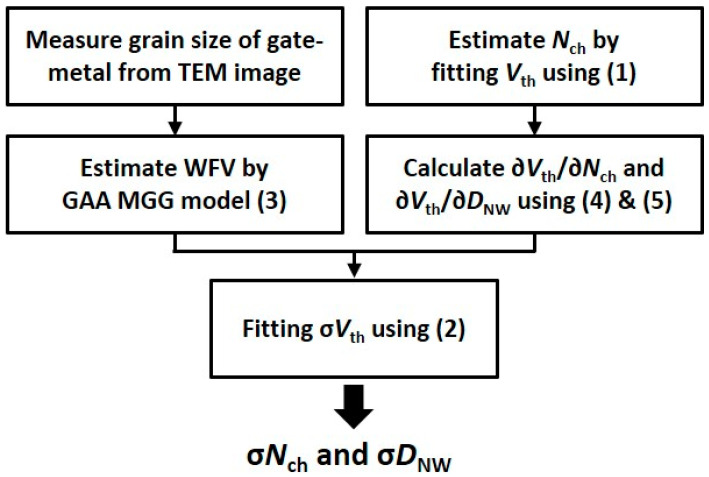
Flowchart of the proposed σ*V*_th_ modeling process.

**Figure 5 nanomaterials-12-01721-f005:**
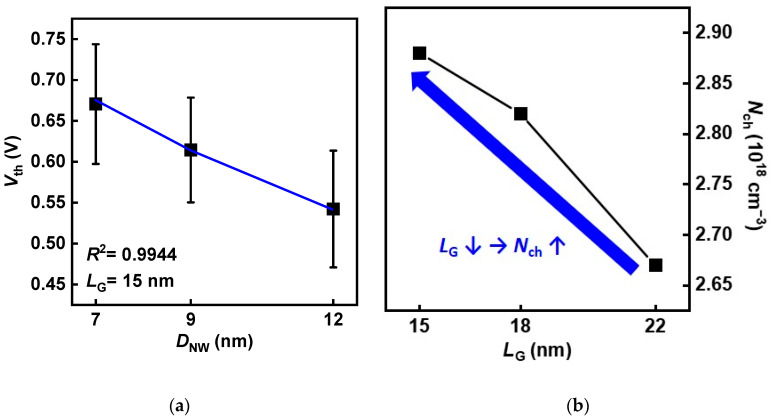
(**a**) Measured (black dots) and modeled (blue line) values of *V*_th_ as a function of D_NW_. (**b**) Extracted *N*_ch_ as a function of *L*_G_.

**Figure 6 nanomaterials-12-01721-f006:**
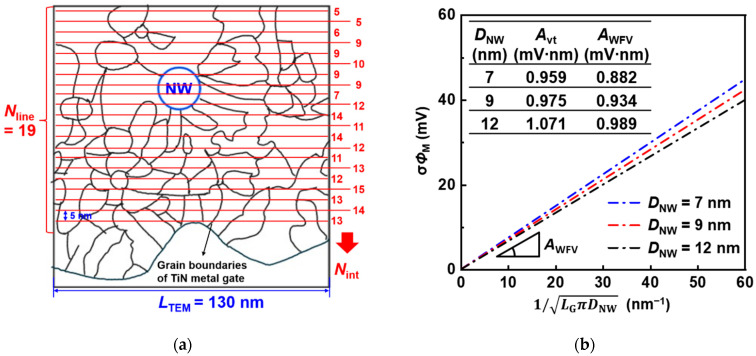
(**a**) Grain boundaries estimated from [19] and red horizontal lines to estimate *G*_size_ of TiN metal gate of SNWFETs. (**b**) Pelgrom’s plot only considering work function variation (WFV) by metal gate granularity (MGG).

**Figure 7 nanomaterials-12-01721-f007:**
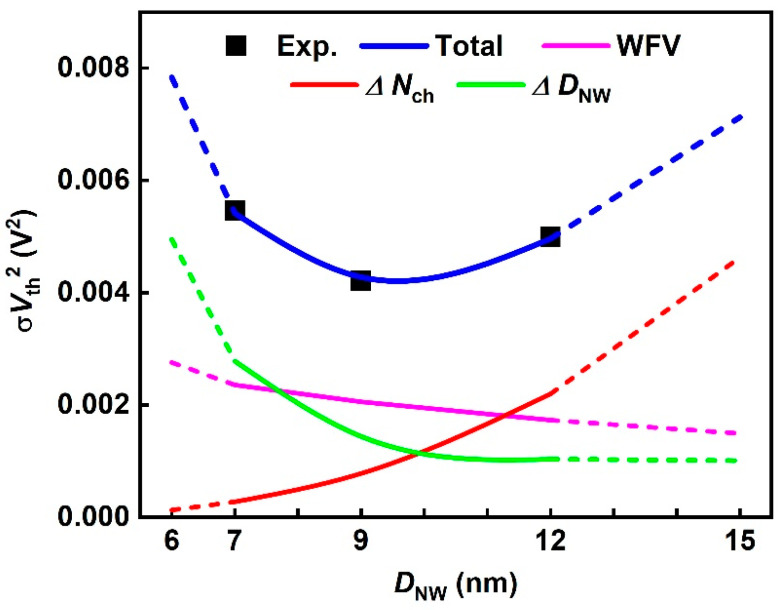
Model fitting results (blue line) considering WFV (pink line), ∆*N*_ch_ (red line), and ∆*D*_NW_ (green line) for the measured value of squared σ*V*_th_ (black dots). The model fits were extrapolated for *D*_NW_ of 6 and 15 nm (dashed line).

**Figure 8 nanomaterials-12-01721-f008:**
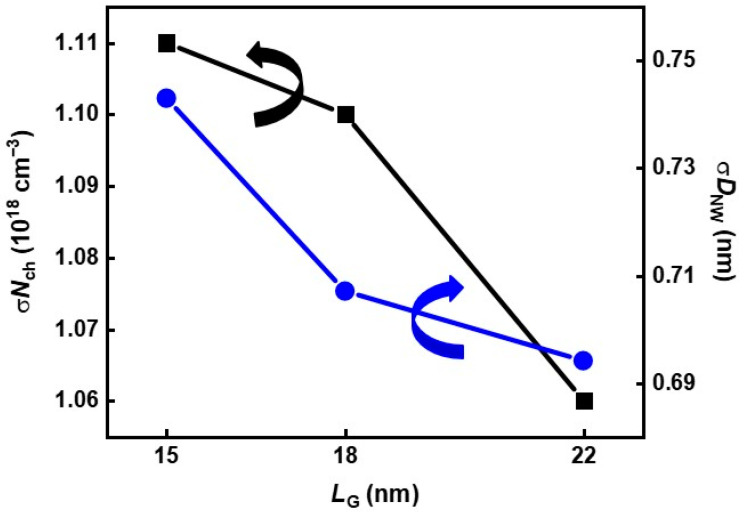
Extracted σ*N*_ch_ (black line) and σ*D*_NW_ (blue line) as function of *L*_G_.

## Data Availability

Not applicable.

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
