# Peer review of "Novel Modeling Approach to Analyze Threshold Voltage Variability in Short Gate-Length (15–22 nm) Nanowire FETs with Various Channel Diameters"

_nanomaterials, 2022, doi:10.3390/nano12101721_

Round 1
Reviewer 1 Report
The authors investigate a model about threshold voltage variation of gate-all-around type silicon-nanowire field-effect transistors. An error propagation law for silicon- nanowire field-effect transistors is newly proposed, which is expressed as a summation of variation of nanowire diameter, variation of channel dopant concentration and metal work function variation. The authors also show an interesting result that dominant variation mechanism depends on a nanowire scale. The model will be useful for development of gate-all-around silicon nanowire field-effect transistors. The paper may be worth publishing in MDPI Nanomaterials.
Author Response
Response to Reviewer 1 Comments
Point 1: The authors investigate a model about threshold voltage variation of gate-all-around type silicon-nanowire field-effect transistors. An error propagation law for silicon- nanowire field-effect transistors is newly proposed, which is expressed as a summation of variation of nanowire diameter, variation of channel dopant concentration and metal work function variation. The authors also show an interesting result that dominant variation mechanism depends on a nanowire scale. The model will be useful for development of gate-all-around silicon nanowire field-effect transistors. The paper may be worth publishing in MDPI Nanomaterials.
Response 1: We appreciate your positive comments, which have supported our manuscript to be published. In the revised manuscript, we have considered all referee comments. Again, thank you for your time and consideration of our manuscript (Nanomaterials 1686663).

Reviewer 2 Report
The authors presented an error propagation method to analyze threshold voltage variation due to several variation sources in SNWFETs. This article is well organized with clear descriptions. Consistent simulation results have been presented and the causes have also been discussed clearly. Its quality is quite well. Only one thing I would like to remind the authors to indicate the full worlds as the abbreviations are used first time in this article, such as MGG on line 67 and RDF on line 72.
I suggest this article can be published as its form.
Author Response
Response to Reviewer 2 Comments
Point 1: The authors presented an error propagation method to analyze threshold voltage variation due to several variation sources in SNWFETs. This article is well organized with clear descriptions. Consistent simulation results have been presented and the causes have also been discussed clearly. Its quality is quite well. Only one thing I would like to remind the authors to indicate the full worlds as the abbreviations are used first time in this article, such as MGG on line 67 and RDF on line 72. I suggest this article can be published as its form.
Response 1: Thank you for your detailed comments. We have double-checked that all abbreviations are written as full words when used for the first time. In the case of MGG and RDF that you pointed out, they were first used on lines 35-36 of the introduction and were written as full words, such as metal gate granularity and random dopant fluctuations, respectively. We also rewrote TCAD and SPICE as technology computer-aided design (line 40) and simulation program with integrated circuit emphasis (line 43), respectively. We appreciate your valuable suggestions, which have helped improve the quality of our manuscript.
Revised contents 1:
- On page 1, 1nd paragraph:
According to previous studies, random variations such as metal gate granularity (MGG), line edge roughness (LER), and random dopant fluctuation (RDF) cause Vth variation in ultrascaled GAA transistors [6-10].
Revised contents 2:
- On page 1, 2nd paragraph:
First, technology computer-aided design (TCAD) simulations are suitable for analyzing the influence of variation sources, but it is difficult to predict the cause of variation inversely from measured Vth variation [6-10]. Second, simulation program with integrated circuit emphasis (SPICE)-based models can be applied to analyze the variation sources of measured Vth variation, but it consumes time and makes an error because all devices should be calibrated [15].
